# Repellent and Toxicant Effects of Eight Essential Oils against the Red Flour Beetle, *Tribolium castaneum* Herbst (Coleoptera: Tenebrionidae)

**DOI:** 10.3390/biology11010003

**Published:** 2021-12-21

**Authors:** El-Said M. Elnabawy, Sabry Hassan, El-Kazafy A. Taha

**Affiliations:** 1Department of Economic Entomology, College of Agriculture, Kafrelsheikh University, Kafrelsheikh 33516, Egypt; elsaid.abdelfatah@agr.kfs.edu.eg; 2Department of Biology, College of Science, Taif University, P.O. Box 11099, Taif 21944, Saudi Arabia; hassan@tu.edu.sa

**Keywords:** essential oils, contact toxicity, *Tribolium castaneum*, repellency effect

## Abstract

**Simple Summary:**

There is a growing need to preserve stored products and cereal grains from loss of weight and quality during storage. *Tribolium castaneum* Herbst is known as a serious pest of stored products. Several studies have estimated the efficacy of essential oils (EOs) against *T. castaneum* but still not enough information on the comparison between repellency and toxicity effects for the EOs. So, this study was to compare the repellency and toxicity effects of eight EOs against the adults of *T. castaneum*. The results indicated that the 5, 10, and 15% concentrations of *Syzygium aromaticum* EO had a higher repellent effect against *T. castaneum* than *A. sativum, E. camaldulensis*, *L. officinalis*, *S. chinensis*, *M. chamomilla*, *C. limon*, and *P. dulcis* after 30 min of exposure. Moreover, the use of *Prunus dulcis* and *Matricaria chamomilla* EOs caused a significantly higher mortality percentage than *Syzygium aromaticum*, *Allium sativum, Eucalyptus camaldulensis*, *Lavandula officinalis*, and *Simmondsia chinensis* at 15% concentration.

**Abstract:**

This study was conducted to compare the repellent effect and contact toxicity of eight essential oils (EOs), including *Syzygium aromaticum*, *Allium sativum, Eucalyptus camaldulensis*, *Lavandula officinalis*, *Simmondsia chinensis*, *Matricaria chamomilla*, *Citrus limon*, and *Prunus dulcis*, against adults of *Tribolium castaneum* Herbst. Four concentrations (1, 5, 10, and 15% in acetone solvent) of each EO were tested. The 5, 10, and 15% concentrations of *S. aromaticum* EO had a high repellency effect against *T. castaneum* compared with *A. sativum*, *E. camaldulensis*, *L. officinalis*, *S. chinensis*, *M. chamomilla*, *C. limon*, and *P. dulcis* after 30 min of exposure. The repellency test of the *S. aromaticum*, *E. camaldulensis*, *L. officinalis*, *M. chamomilla*, *C. limon*, and *P. dulcis* EOs on *T. castaneum* has shown that the mortality percentages enhanced with the increase in the EOs concentration and also with the exposure time. The 15% concentration of *P. dulcis* and *M. chamomilla* EOs have a significant impact on the mortality rate of *T. castaneum* compared with *S. aromaticum*, *A. sativum, E. camaldulensis*, *L. officinalis*, and *S. chinensis* after the 24 h of contact test. Moreover, the 15% concentration of the *C. limon* EO caused a greater mortality percentage compared with *S. aromaticum*, *A. sativum, E. camaldulensis*, and *L. officinalis*. It could be concluded that using the *S. aromaticum* EO as a repellent oil and using *P. dulcis, M. chamomilla*, and *C. limon* for contact toxicity to treat the flour infested by *T. castaneum* can play an important role in protecting stored grains and their products.

## 1. Introduction

A large amount of food loss during storage due to pest infestation is a real problem in developed and developing countries, resulting in large financial losses [1,2]. Some species of insects, mites, and fungi feed on stored grains, degrading product quality and causing net losses of 9 to 20% [3]. About 1660 species of insects are known to have an impact on the quality of preserved foods [4,5]. *Tribolium castaneum* Herbst is one of the most common insect pests found in stored grains and it is only able to feed on the grains that have already been attacked by the primary pests. The infestation with *T. castaneum* has a direct effect on the quantitative and qualitative properties of the stored products [6].

Conventional contact insecticides have mostly controlled stored goods pests since 1960 [7,8]. The use of such insecticides is increasingly being challenged more and more. Insects become resistant to those pesticides; additionally, the risk of the residues of those chemical pesticides cause harmful effects to the environment and human health, which resulted in the use of those compounds being increasingly restricted [8,9,10]. The demand for food safety and a pollutant-free environment has increased, highlighting the need for safe alternative control methods [10,11].

Plant EOs are natural components extracted from different plant parts with insecticidal properties for plant protection to avoid the side effects of synthetic chemical insecticides. They have different modes of action and chemical properties which can reduce the insect pest populations in various ways, particularly toxicants, repellents, antifeedants, and attractants. The role of plant EOs as effective insecticides has been studied with references to different insect pests [12,13].

Essential oils, in particular, have some intriguing qualities that could make them a viable alternative to synthetic insecticides [14,15]. Because of their unique qualities, EOs are becoming more popular as a pesticide alternative [16]. The different EOs are less persistent in the environment than the conventional pesticides because of their extreme volatility, temperature, and Ultraviolet light destruction sensitivity [17]. Furthermore, when compared with conventional insecticides, many EOs have low toxicity on the mammalian and are environmentally friendly [18]. Previous research has indicated that EOs of four spice plants and their main components have clear toxicity and repellant actions against *T. castaneum* and *Lasioderma serricorne* [19]. Jahromi et al. [19] have indicated that, at high concentrations, the natural garlic emulsion has the greatest repellency impact against *T. castaneum*. Moreover, *Oryzaephilus surinamensis* has completely died by the essential oil of *M. chamomilla* at concentrations greater than 0.5% [20]. At a concentration of 0.5%, lemongrass EO is extremely repellent to *Acanthoscelides obtectus* and *T. castaneum* [21].

Some studies have demonstrated that some monoterpenoids, constituents of EOs, affect many insect pests [22,23]. Jiang et al. [24] have indicated that the linalool component has repellent and insecticidal activities. The linalool component has been reported as an acetylcholinesterase inhibitor [25]. Moreover, the EOs insecticidal efficacy has been shown by Amy et al. [26], who have reported that the linalool component inhibits both c-aminobutyric acid type A receptors and nicotinic acetylcholine receptors.

In recent years, several previous researchers have reported the repellency or mortality effects of *Syzygium aromaticum*, *Allium sativum*, *Eucalyptus camaldulensis*, *Lavandula officinalis*, *Simmondsia chinensis*, *Matricaria chamomilla*, *Citrus limon*, and *Prunus dulcis* EOs, against *T. castaneum*. However, detailed information about the comparison between the repellency and toxicity effects is still lacking. So, the main aim of this study was to compare between repellency and toxicity effects of the eight tested EOs to determine the best application.

## 2. Materials and Methods

### 2.1. Experimental Site

The experiment was conducted in the laboratory of the Economic Entomology Department, College of Agriculture, Kafrelsheikh University, Kafrelsheikh, Egypt. 

### 2.2. Essential Oils

The eight EOs, clove (*Syzygium aromaticum* L.), garlic (*Allium sativum* L.), river red gum (*Eucalyptus camaldulensis* Dehnh), lavender (*Lavandula officinalis* L.), jojoba (*Simmondsia chinensis* (Link) C. Schneid), chamomile (*Matricaria chamomilla* L.), lemon (*Citrus limon* Burm), and almond (*Prunus dulcis* Mill. D. A. Webb) were purchased from El Captain Company for EOs, Egypt (www.elcaptain.net (accessed on 19 December 2021)). 

### 2.3. The Red Flour Beetle (T. castaneum)

The unsexed adults of *T. castaneum* were collected from the wheat grain stores and brought to the laboratory of the Economic Entomology Department, College of Agriculture, Kafrelsheikh University, Egypt, for rearing. The *T. castaneum* was reared in open plastic jars covered with a muslin cloth. The jars were kept in the laboratory for three months for rearing. Then 200 adults were transferred to a new plastic jar with 50 g of fresh wheat flour and the adults were kept for 4 days for laying eggs. Then, they were removed and the plastic jar was kept at 27 ± 2 °C and 65 ± 5% R.H. The progenies of these stocks were nearly identical in size and age.

### 2.4. Repellency Assay 

The EOs were tested for repellency effect for *T. castaneum* in glass Petri dishes (10 cm diam). The used dilution percentages of the EOs were 1, 5, 10, and 15% (*v*/*v*), which were prepared in an acetone solvent to obtain the required solutions. Whatman filter papers (9 cm diam) were cut into two equal parts. A total 0.5 mL of each oil percentage was uniformly applied to half of the filter paper using a pipette and the second half of filter paper was treated with acetone only. Both treated and untreated pieces were placed in a Petri dish, and they were kept for one h to evaporate the solvent. Then, 30 adult individuals of *T. castaneum* were inserted in the Petri dish between the two halves of the filter paper, and then the Petri dishes were covered. Each EO percentage was repeated in five replicates. The total numbers of *T. castaneum* on treated and untreated halves were counted and recorded after 30 and 180 min and the repellency percentages were conducted using this equation [27]: PR = [(Nc − Nt)/(Nc + Nt)] × 100
where PR means the repellency percentages after exposure time; Nc means the insect numbers on the untreated area after the exposure; Nt means the insect numbers on the treated area after the exposure.

### 2.5. Insect Mortality Bioassay (Contact Toxicity)

The direct effect of the EOs toxicity towards adults of *T. castaneum* was assessed using a direct contact assay [28]. In acetone solvent, the different concentrations (1, 5, 10, and 15% (*v*/*v*) of the tested EOs were prepared. One ml of each concentration was applied in a glass Petri dish (10 cm diam) using a pipette. It was left for two h until the solvent evaporated. After that, 30 adults of *T. castaneum* were added to each Petri dish separately. The dishes with solvent only were used as a control treatment. Each treatment was repeated in five replicates. After 24 h of treatment, the numbers of dead insects were recorded, and the mortality percentages were calculated. The values of LC_50_ and LC_90_ as well as their confidence limits were conducted by using probit analysis by SPSS software [29] according to Finney’s method [30].

### 2.6. Statistical Analysis

The experimental data were analyzed in univariate analysis using SPSS software [29]. The Shapiro–Wilk normality test was used to test the normality of the data, which indicated the normal distribution of the data. Therefore, the analysis was performed on the original data. The statistical analysis of data was conducted on each dependent variable and the experimental treatments were compared for significant differences with a two-way ANOVA and the differences between the means were estimated using Tukey’s test.

## 3. Results

### 3.1. Repellency Assay

The repellency effect of 1% (*v*/*v*) of *S. aromaticum* EO against *T. castaneum* was greater than *E. camaldulensis*, *L. officinalis*, *S. chinensis*, *M. chamomilla*, *C. limon*, and *P. dulcis* after 30 min of exposure time. The repellent actions of 5, 10, and 15% (*v*/*v*) of *S. aromaticum* oil on *T. castaneum* were significantly higher than *A. sativum*, *E. camaldulensis*, *L. officinalis*, *S. chinensis*, *M. chamomilla*, *C. limon*, and *P. dulcis* after 30 min of exposure time (Table 1). Moreover, the EO of *A. sativum* at 15% concentration had a higher significant repellency effect compared with *E. camaldulensis*, *L. officinalis*, *S. chinensis*, *C. limon*, and *M. chamomilla* after 30 min of exposure time. The 1 and 5% of *S. aromaticum* EO had higher repellency effects against the adults of *T. castaneum* than *A. sativum*, *E. camaldulensis*, *S. chinensis*, *M. chamomilla*, *C. limon*, and *P. dulcis* after 180 min of exposure (Table 1). The 15% of *S. aromaticum* oil was more repellent than *E. camaldulensis*, *L. officinalis*, *S. chinensis*, *C. limon*, and *M. chamomilla* after 180 min of exposure (Table 1).

Furthermore, Table 2 shows that the F values of the concentrations were statistically significant for *S. aromaticum*, *A. sativum*, *P. dulcis*, *L. officinalis*, *E. camaldulensis*, *C. limon*, and *M. chamomilla* EOs. The F values of the exposure time were statistically significant for the EOs of *S. aromaticum*, *P. dulcis*, *L. officinalis*, *S. chinensis*, *E. camaldulensis*, *C. limon*, and *M. chamomilla*. Moreover, the F values of the interaction between concentration and exposure time were significant for *S. aromaticum, A. sativum*, *C. limon*, and *M. chamomilla* EOs (Table 2).

### 3.2. Insect Mortality Bioassay (Contact Toxicity)

The results of the present study indicated that *P. dulcis* EO had a high performance against *T. castaneum* after 24 h of exposure time. The LC_50_ value of *P. dulcis* was 6.62%, followed by *M. chamomilla* with an LC_50_ value of 7.78%, while the LC_50_ values of *L. officinalis* and *S. aromaticum* EOs were 23.58 and 17.58%, respectively, after 24 h from the treatment of *T. castaneum* (Table 3). In addition, the order of tested EOs efficacy against adults of red flour beetle based on LC_90_ values followed the same trend of almost LC_50_ values. The LC_90_ values of *P. dulcis*, *M. chamomilla*, and *C. limon* were 16.11, 16.13, and 16.54%, respectively, while the LC_90_ values of *L. officinalis, E. camaldulensis*, and *S. aromaticum* EOs were 38.64, 34.82, and 30.38%, respectively, after 24 h from the treatment of *T. castaneum* (Table 3).

As shown in Figure 1, 15% concentrations of *P. dulcis* and *M. chamomilla* EOs have higher mortality rates of *T. castaneum* than those of *S. aromaticum, A. sativum, E. camaldulensis*, *L. officinalis*, and *S. chinensis* after the 24 h of contact test. Moreover, *C. limon* EO had a higher mortality rate than those of *S. aromaticum, A. sativum, E. camaldulensis*, and *L. officinalis* in 15% of concentration. The mortality percentages of *P. dulcis*, *M. chamomilla*, and *C. limon* were 81.33, 81.33, and 77.33%, respectively, in 15% concentration. The mortality rates were significantly higher in 10 and 15% concentrations compared with 1 and 5% concentrations of *P. dulcis*, *M. chamomilla*, and *C. limon* EOs. So, the mortality percentages of *T. castaneum* are increased by increasing the EOs concentration.

## 4. Discussion

### 4.1. Repellent Effect 

Recently, many researchers have evaluated the repellency or mortality effects for the tested EOs. However, detailed results about the comparison between the repellent and insecticidal effects are still insufficient. Thus, this study was conducted to find out the differences between repellency and toxicity effects of the tested EOs.

In this study, the repellence tests indicated that *S. aromaticum* and *A. sativum* EOs had good potential as repellent substances against *T. castaneum* after 30 min of exposure. The effectiveness of *S. aromaticum* and *A. sativum* EOs could be due to some factors which contribute to enhancing the repellent activity, such as the odour, chemical compositions, and main components, as shown in previous works, such as Abo-El-Saad [31] who has shown that the significant insecticidal activity of clove oil might be attributed to the main components, eugenol and β-caryophyllene. The high repellent activity of *S. aromaticum* EO may be due to eugenol because it is the main constituent of clove oil (48.92%) as determined by GC-MS [31]. Moreover, Boraei [32] reported that the main component of clove EO was eugenol (37.43%). In many previous studies, this component has been indicated to be a strong repellent compound against many insect pests [33,34]. Moreover, the results indicated that the 15% of *A. sativum* EO had a higher repellency effect than *E. camaldulensis*, *L. officinalis*, *S. chinensis*, *C. limon*, and *M. chamomilla*. These findings are in harmony with those reported by Jahromi et al. [19], where they have shown that the natural garlic emulsion, *A. sativum*, had the highest repellency effect against *T. castaneum* at the high concentration. In addition, the *T. castaneum* was more susceptible to garlic emulsion than *Lasioderma serricorne* [19]. Rahman and Motoyama [35] found that the active volatile components are sulfide components made by the rapid degradation of allicin. Rahman and Schmidt [36] have studied the chemical analysis of garlic extract by GC-MS and have indicated that allicin was a main component and had repellency impacts on some insect pests. Our results showed that 1% concentration of *M. chamomilla* EO repelled 2.22% of *T. castaneum* after 30 min of exposure while Al-Jabr [20] has shown that the repellent effect of *M. chamomilla* against *T. castaneum* was 84.73% after 48 h at 1% concentration. The differences between our findings and Al-Jabr [20] may be because Al-Jabr [20] has used a long exposure time (48 h). In addition, the analysis of data revealed that the repellency assay of the *S. aromaticum, E. camaldulensis*, *L. officinalis*, *M.chamomilla*, *C. limon*, and *P. dulcis* EOs on *T. castaneum* has revealed that the repellency percentages enhanced with the increase in the EOs concentration and also with the exposure time. (Table 2). These findings are in harmony with those that have been indicated by Jazia et al. [37]; they have reported that the repellent impact of coriander EO is highly dependent upon EO concentration and exposure period. We suppose the reason for the low mortality percentage of *S. aromaticum* by direct contact may be due to its low toxicant components which are attached to the insect body. These results may recommend that using *S. aromaticum* as a repellent EO and *P. dulcis* and *M. chamomilla* EOs for a contact toxicity might be a promising method to control *T. castaneum*.

### 4.2. Toxicity Effect (Contact Toxicity)

The results indicated that a 15% concentration of *P. dulcis* and *M. chamomilla* were more effective by contact toxicity against *T. castaneum* than *S. aromaticum, A. sativum, E. camaldulensis*, *L. officinalis*, and *S. chinensis*. Moreover, a 15% concentration of *C. limon* EO had a higher mortality rate than *S. aromaticum, A. sativum, E. camaldulensis*, and *L. officinalis*. The effectiveness of *P. dulcis* and *M. chamomilla* are in harmony with those that have been reported by Al-Jabr [20], who has shown that 1% concentration of *P. amygdalus* was more effective against *T. castaneum* with complete mortality after 14 days of exposure compared with *Cinnamomum camphora, Cymbopogon winterianus, M. chamomilla, Mentha viridis, P. amygdalus* var *amara, Rosmarinus afficinalis*, and *S. chinensis*. Azab et al. [38] have shown that the LC_50_ and LC_90_ of sweet almond EO to the adults of *O. surinamensis* were 4.52 and 5.55% (v/w), respectively, after 7 days of exposure time and their mortality percentages are enhanced by increasing the EOs concentration and the period of exposure. Moreover, Matsumoto et al. [39] have shown that the EOs of bitter almond, spearmint, and birch bark were used in a composition that was sold as a pesticide, insect repellent, and acaricide. Al-Jabr [20] has indicated that *M. chamomilla* EO at a concentration of more than 0.5% had a complete mortality effect against *O. surinamensis*. Moreover, Padin et al. [40] have shown that the methanolic extracts of *M. chamomilla* had a 57% mortality rate against *T. castaneum* after 7 days. El-Bakry et al. [41] have shown that the LC_50_ value of *C. sinensis* was 35 µL/L and it was the most effective one against *T. castaneum*. 

Generally, the major components of plant EOs are many monoterpenoids, such as d-limonene, Æ-terpineol, â-myrcene, pulegone, and linalool, which affect negatively on many insect pests such as the German cockroach and the house fly [22,23]. Linalool was considered as an acetylcholinesterase inhibitor and it has been demonstrated as a potent contributor to the repellent and insecticidal activities [24,25]. Bhavaniramya et al. [42] have indicated that the use of EOs of clove, lemon, thyme, and cinnamon have increased the storage periods and have kept the good quality of food safety. Plant EOs are generally utilized in food products for food preservation because of their odour, tastes, and strong antibacterial properties. They contain terpenes and aromatic volatile chemicals, which play a significant role in food safety without decreasing quality [41]. For instance, the EOs of citrus, including monoterpenes, sesquiterpenes, and oxygenated derivatives, have strong inhibitory effects against harmful bacteria, proposing that they could be used as flavouring and antioxidants preservative materials of food [42]. Thus, the use of EOs can protect the food products for a long time without decreasing the quality. 

This study may suggest that using *S. aromaticum* as a repellent material and *P. dulcis* and *M. chamomilla* EOs for contact mortality might be useful for the management of *T. castaneum* to protect stored grains and their products. However, in the future, investigations are needed to compare the repellency and toxicity effects of tested EOs against other stages of *T. castaneum* Herbst, including egg, larvae, and pupae. Moreover, in the future, we would like to evaluate the acetone residues on treated stored grains and their products to identify the level of safety.

## Figures and Tables

**Figure 1 biology-11-00003-f001:**
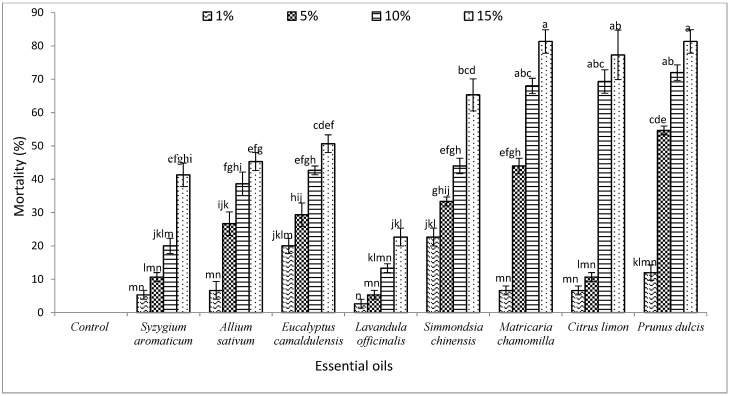
The impact of eight essential oils on the mortality rate of *T. castaneum* (Herbst) after 24 h of exposure. Different letters above the bars indicate a significant (*p* < 0.05) difference.

**Table 1 biology-11-00003-t001:** Repellency impact of eight essential oils against *T. castaneum* after 30 and 180 min of exposure.

Essential Oils	Repelled Adults (%) after 30 min of Exposure				Repelled Adults (%) after 180 min of Exposure			
	Concentrations				Concentrations			
	1%	5%	10%	15%	1%	5%	10%	15%
*S. aromaticum*	64.44 a	77.77a	82.22 a	95.55 a	82.22 a	86.66 a	95.55 a	100.00 a
*A. sativum*	51.11ab	51.11 b	60.00 b	73.33 b	55.55 bc	60.00 b	64.44 b	91.11 ab
*S. chinensis*	40.00 bc	48.88 b	31.11 cd	31.11 de	60.00 bc	64.44 b	73.33 b	73.33 c
*P. dulcis*	22.22 cd	44.44 b	53.33 b	57.77 bc	42.22 c	64.44 b	82.22 ab	91.11 ab
*L. officinalis*	17.77 d	31.11 bc	48.88 bc	48.88 cd	73.33 ab	73.33 ab	77.77 ab	82.22 bc
*E. camaldulensis*	8.88 d	26.66 bc	44.44 bc	44.44 cd	60.00 bc	64.44 b	64.44 b	77.77 bc
*C. limon*	6.66 d	22.22 bc	57.77 b	31.11 de	2.22 d	55.55 b	77.77ab	82.22 bc
*M. chamomilla*	2.22 d	4.44 c	22.22 d	22.22 e	0.00 d	15.55 c	46.66 c	66.66 c

Values are the mean percentages of repelled insects. The means of each column followed by the same letter do not differ significantly *p* > 0.05 as determined by Tukey’s test.

**Table 2 biology-11-00003-t002:** Analysis of variance of different EOs of the tested insect’s repellency.

Essential Oils	Concentrations				Exposure Time				Interaction between Concentrations and Exposure Time			
	df	MS	F Value	*p*-Value	df	MS	F Value	*p*-Value	df	MS	F Value	*p*-Value
*S. aromaticum*	3	2200.00	17.47	0.000 **	1	1896.30	15.06	0.001 **	3	444.44	3.53	0.039 *
*A. sativum*	3	1037.04	8.24	0.002 **	1	474.07	3.76	0.070 ^NS^	3	59.26	0.47	0.707 ^NS^
*S. chinensis*	3	46.91	0.20	0.892 ^NS^	1	5400.00	23.52	0.000 **	3	46.91	0.20	0.892 ^NS^
*P. dulcis*	3	2081.48	17.56	0.000 **	1	3918.52	33.06	0.000 **	3	66.67	0.56	0.647 ^NS^
*L. officinalis*	3	533.33	4.00	0.027 *	1	9600.00	72.00	0.000 **	3	207.41	1.56	0.239 ^NS^
*E. camaldulensis*	3	800.00	9.00	0.001 **	1	7585.18	85.33	0.000 **	3	246.91	2.78	0.075 ^NS^
*C. limon*	3	4717.90	32.25	0.000 **	1	3750.00	25.63	0.000 **	3	703.09	4.81	0.014 *
*M. chamomilla*	3	2200.00	17.47	0.000 **	1	1896.30	15.06	0.001 **	3	444.44	3.53	0.039 *

* indicates *p* < 0.05, ** indicates *p* < 0.01, ^NS^ indicates *p* > 0.05.

**Table 3 biology-11-00003-t003:** The contact toxicity of the eight essential oils against the adult stage of *T. castaneum* after 24 h from exposure time.

Essential Oil	LC_50_ (%)	Confidence Limits 95%		LC_90_ (%)	Confidence Limits 95%	
		Lower	Upper		Lower	Upper
*Lavandula officinalis*	23.58	18.80	36.34	38.64	29.20	64.91
*Syzygium aromaticum*	17.58	15.02	22.33	30.38	24.81	41.67
*Allium sativum*	14.90	12.57	19.13	30.22	24.25	42.86
*Eucalyptus camaldulensis*	14.02	11.24	19.97	34.82	26.17	57.95
*Simmondsia chinensis*	10.73	8.84	13.41	26.90	21.65	37.76
*Citrus limon*	9.59	8.02	11.35	16.54	14.19	20.74
*Matricaria chamomilla*	7.78	6.72	8.84	16.13	14.39	18.66
*Prunus dulcis*	6.62	5.37	7.77	16.11	14.17	19.05

## Data Availability

All data generated or analyzed during this study are included in this published article.

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
