# Peer review of "Repellent and Toxicant Effects of Eight Essential Oils against the Red Flour Beetle, Tribolium castaneum Herbst (Coleoptera: Tenebrionidae)"

_biology, 2021, doi:10.3390/biology11010003_

Round 1

Reviewer 1 Report

  • In my opinion the  Simple Summary and the abstract is repetition, it’s better to give one and remove the second
  • Write scientific names (plant species and insects ) in italic in all text
  • Four concentrations (1, 5, 10, and 15% in acetone solvent) of each essential oil were: this is percentage not concentrations
  • The essential oils of dulcis, M. chamomilla, and C. lemon have a significant impact on the mortality percentage
  • The major constituents of essential oils are monoterpenoids which affect many insect pests [19,20]. Also, the linalool component has been reported as an acetylcholinesterase inhibitor [21]. In my opinion it’s better to change these two sentences like this : Some studies have demonstrated that some monoterpenoids, constituents of essential oils affect many insect pests [19,20]. Also, the linalool component has been reported as an acetylcholinesterase inhibitor [21].
  • Also, the linalool component has been reported as an acetylcholinesterase inhibitor [21]. Indicate the relation between this activity and repellency activity 
  • Remove : as a safe alternative control method from the last sentence in introduction
  • Add reference(s) to Repellency assay
  • I believe it’s better to write this sentence like this: The essential oils, aromaticum, A. sativum, E. camaldulensis, L. officinalis, S. chinensis, M. chamomilla, C. lemon, and P. dulcis were effective against T. castaneum and significantly varied depending on the botanical origin of the essential oil and concentration.
  • You said that: The repellency impact of the essential oil of aromaticum was significantly higher than all tested essential oils except A. sativum. ………….and in other sentence………….. The essential oil of A. sativum ranked in second place and ………..where is the correct idea?
  • Discussion Repellent effect: In this study, the repellence tests indicated …….. tested essential oils. Also, the results support …….. with the high concentration. Also, aromaticum oil had …. of T. castaneum even with the low concentration. You said also….also….. change one
  • In discussion, Azab et al. [33] have shown that the mortality percentage of the almond oil on the adults of Oryzaephilus surinamensis was 98.0% at 5% (v/w) concentration after 21 days of exposure. Also, Matsumoto et al. [34] have shown that essential oil of bitter almond was combined into a mixture having insecticide, acaricide, and repellent characteristics. In these two studies researchers have done their work with essential oils of different plants and on different insects, which I see it incomparable

Author Response

Reviewer 1, comments and Suggestions for Authors

The authors would like to thank you for your valuable and great comments. It was your valuable and insightful comments that led to possible improvements in the current version. The corrections were done according to your advice.

Comment 1: In my opinion the Simple Summary and the abstract is repetition, it’s better to give one and remove the second.

Response 1: Thank you very much for your great comment we changed the simple summary according to your advice and it becomes different from the abstract.

Comment 2: Write scientific names (plant species and insects ) in italic in all text

Response 1: Thank you very much for your great advice: All scientific names were changed to Italic.

Comment 3: Four concentrations (1, 5, 10, and 15% in acetone solvent) of each essential oil were: this is percentage not concentrations.

Response 3: Thank you very much for your comment: In this study we used percentages as a one form of concentration if you mean we have to use another concentration form such as ppm we can convert the used percentage to ppm by (Percentage multiplied by 104). So we can do it if you recommend us to use ppm.

Comment 4: The essential oils of dulcisM. chamomilla, and C. lemon have a significant impact on the mortality percentage

Response 4: Thank you very much for your suggestion and the sentence was changed. As shown in Fig. (1), 15 % concentrations of P. dulcis and M. chamomilla EOs have higher mortality rates of T. castaneum than those of S. aromaticum, A. sativum, E. camaldulensis, L. officinalis, and S. chinensis after 24 hrs of contact test.

Comment 5: The major constituents of essential oils are monoterpenoids which affect many insect pests [19,20]. Also, the linalool component has been reported as an acetylcholinesterase inhibitor [21]. In my opinion it’s better to change these two sentences like this : Some studies have demonstrated that some monoterpenoids, constituents of essential oils affect many insect pests [19,20]. Also, the linalool component has been reported as an acetylcholinesterase inhibitor [21].

Response 5: Thank you very much for your comment we rechecked the sentence and changed it according to your advice: Some studies have demonstrated that some monoterpenoids, constituents of EOs affect many insect pests [22,23]. Jiang et al. [24] have indicated that the linalool component has repellent and insecticidal activities.

Comment 6: Also, the linalool component has been reported as an acetylcholinesterase inhibitor [21]. Indicate the relation between this activity and repellency activity

Response 6: Thank you very much for your comment we revised the sentence and added some references to indicate that. Some studies have demonstrated that some monoterpenoids, constituents of EOs affect many insect pests. Jiang et al. [24] indicated that the linalool component has repellent and insecticidal activities. The linalool component has been reported as an acetylcholinesterase inhibitor [25]. Also the EOs insecticidal efficacy was shown by Amy et al. [26], who reported that linalool component inhibits both of c-aminobutyric acid type A receptors (GABAARs) and nicotinic acetylcholine receptors (nAChR).

Comment 7: Remove: as a safe alternative control method from the last sentence in introduction Response 7: Thank you very much for your comment we removed it: The sentence was removed.

Comment 8: Add reference(s) to Repellency assay

Response 8: Thank you very much for your comment we revised the manuscript and we added some differences according to your advice: Previous research indicated that EOs of four spice plants and their main components have clear toxicity and repellant actions against T. castaneum and Lasioderma serricorne [19]. Jahromi et al. [19] indicated that at high concentrations, the natural garlic emulsion has the greatest repellency impact against T. castaneum. Also, Oryzaephilus surinamensis was completely died by the essential oil of M. chamomilla at concentrations greater than 0.5 percent [20]. At a concentration of 0.5 percent, lemongrass essential oil was found to be extremely repellent to Acanthoscelides obtectus and T. castaneum as indicated by Gvozdenac et al. [21].

Comment 9: I believe it’s better to write this sentence like this: The essential oils, aromaticum, A. sativum, E. camaldulensis, L. officinalis, S. chinensisM. chamomilla, C. lemon, and P. dulcis were effective against T. castaneum and significantly varied depending on the botanical origin of the essential oil and concentration.

Response 9: Thank you very much for your comment we revised the manuscript and changed it according to your advice: The eight EOs, clove (Syzygium aromaticum L.), garlic (Allium sativum L.), river red gum (Eucalyptus camaldulensis Dehnh), lavender (Lavandula officinalis L.), jojoba (Simmondsia chinensis (Link) C. Schneid), chamomile (Matricaria chamomilla L.), lemon (Citrus lemon Burm), and almond (Prunus dulcis Mill. D. A. Webb).

Comment 10: You said that: The repellency impact of the essential oil of aromaticum was significantly higher than all tested essential oils except A. sativum. ………….and in other sentence………….. The essential oil of A. sativum ranked in second place and ………..where is the correct idea?

Response 10: Thank you very much for your comment, we revised the manuscript and we corrected it according to your advice and Table 1:  The repellency effect of 10 and 15% (v/v) of S. aromaticum oil on T. castaneum was significantly higher than E. camaldulensis, L. officinalis, S. chinensis, M. chamomilla, C. lemon, and P. dulcis (Table 1). Also, the oil of A. sativum at 15% concentration had a higher significant repellency effect compared with E. camaldulensis, L. officinalis, S. chinensis, and M. chamomilla.

Comment 11: Discussion Repellent effect: In this study, the repellence tests indicated …….. tested essential oils. Also, the results support …….. with the high concentration. Also, aromaticum oil had …. of T. castaneum even with the low concentration. You said also….also….. change one

Response 11: Thank you very much for your comment we revised the sentence and changed it according to your advice: In this study, the repellence tests indicated that S. aromaticum and A. sativum EOs had good potential as repellent substances against T. castaneum after 30 min of exposure. Also, the results support that the plant EOs increase in repellency with a high percentage.

Comment 12: In discussion, Azab et al. [33] have shown that the mortality percentage of the almond oil on the adults of Oryzaephilus surinamensis was 98.0% at 5% (v/w) concentration after 21 days of exposure. Also, Matsumoto et al. [34] have shown that essential oil of bitter almond was combined into a mixture having insecticide, acaricide, and repellent characteristics. In these two studies researchers have done their work with essential oils of different plants and on different insects, which I see it incomparable.

Response 12: Thank you very much for your comment, I rechecked the findings and I corrected it: Azab et al. [36] have shown that the LC50 and LC90 of sweet almond EO to adults of Oryzaephilus surinamensis were 4.52% and 5.55% (v/w), respectively after 7 days of exposure time and their mortality percentages are enhanced by increasing the EOs percentage and the period of exposure. Also, Matsumoto et al. [37] have shown that the essential oils of bitter almond, spearmint and birch bark were used in a composition that was sold as a pesticide, insect repellent and acaricide.

Reviewer 2 Report

The manuscript “Repellent and toxicant effects of eight essential oils against the red flour beetle, Tribolium castaneum Herbst (Coleoptera: Tenebrionidae) ”  evaluated the repellent effect and contact toxicity of eight essential oils against adults of the red flour beetle, and it was concluded that using the essential oils to treat the flour infested by T. castaneum can play an important role in the integrated pest management program for protecting stored grains and their products. However, there are still the following problems in the article, it is recommended to major revision:

  1. In section "Repellency assay ", "which were prepared in an acetone  solvent to obtain the required solutions. "The essential oil solution is prepared with acetone, so how to solve the toxicity to the human body when used in actual food?
  2. After using the several essential oils mentioned in the article for application, please explain how it will affect the quality of food, including color, flavor and other characteristics.
  3. Which data can show that your essential oil has a higher mortality rate after use, I have not seen it, please point out.In addition, in Table 1, I only saw that after using the same concentration of essential oils, after a period of time, the quantity increased, and it did not achieve the repellent effect you mentioned.
  4. There are many format and grammatical errors in the manuscript, please modify the full manuscript.
  5. Which paragraph of text corresponds to your picture 1? How was the data in Table 2 obtained?

Author Response

Reviewer 2, comments and Suggestions for Authors

The authors would like to thank you for your valuable and great comments. It was your valuable and insightful comments that led to possible improvements in the current version. All corrections were done according to your advices.

Comment 1: In section "Repellency assay ", "which were prepared in an acetone solvent to obtain the required solutions. "The essential oil solution is prepared with acetone, so how to solve the toxicity to the human body when used in actual food?

Response 1:  Thank you very much for your great comment: Regarding the toxicity of acetone, acetone is used as a solvent, it is easy to evaporate under room temperature so after a short time of storage the stored product will be free of acetone.

Comment 2: After using the several essential oils mentioned in the article for application, please explain how it will affect the quality of food, including color, flavor and other characteristics.

Response 2:  Thank you very much for your great comment, according to your comment we added some references to explain that point: Bhavaniramya et al. [41] have indicated that the use of EOs, clove, lemon, thyme, and cinnamon have increased the storage periods and have kept the good quality of food safety. Plant EOs are generally utilized in food products for food preservation because of their odour, tastes, and strong antibacterial properties. For instance, the EOs of citrus including monoterpenes, sesquiterpenes, and oxygenated derivatives have strong inhibitory effects against harmful bacteria, proposing that they could be used as flavouring and antioxidants preservative materials of food [41]. Almost EOs contain terpenes and aromatic volatile chemicals, which play a significant role in food safety without decreasing quality [41]. Thus, the use of EOs can protect the food products for a long time without decreasing the quality.

Comment 3: Which data can show that your essential oil has a higher mortality rate after use, I have not seen it, please point out. In addition, in Table 1, I only saw that after using the same concentration of essential oils, after a period of time, the quantity increased, and it did not achieve the repellent effect you mentioned.

Response 3: Thank you very much for your questions. We added the differences in Fig. 1 to show high mortality rates and the differences between the EOs and we showed that in the following paragraph. Also, we rechecked the statistical analysis and we added the differences as shown in Table 1 then we illustrated all changes in the second following paragraph.

As shown in Fig. (1), 15 % concentrations of P. dulcis and M. chamomilla EOs have higher mortality rates of T. castaneum than those of S. aromaticum, A. sativum, E. camaldulensis, L. officinalis, and S. chinensis after 24 hrs of contact test. Also, C. lemon EO had a higher mortality rate than those of S. aromaticum, A. sativum, E. camaldulensis, and L. officinalis in 15% of concentration. The mortality percentages of P. dulcis, M. chamomilla, and C. lemon were 81.33, 81.33, and 77.33%, respectively, in 15% concentration. The mortality rates were significantly higher in 10 and 15% concentrations compared with 1 and 5% concentrations of P. dulcis, M. chamomilla, and C. lemon EOs. So, the mortality percentages in some of the tested EOs are increased by increasing the EOs concentration.

The repellency effect of 10 and 15% (v/v) of S. aromaticum oil on T. castaneum was significantly higher than E. camaldulensis, L. officinalis, S. chinensis, M. chamomilla, C. lemon, and P. dulcis (Table 1). Also, the oil of A. sativum at 15% concentration had a higher significant repellency effect compared with E. camaldulensis, L. officinalis, S. chinensis, and M. chamomilla. The 15% of S. aromaticum, P. dulcis, E. camaldulensis, and C. lemon EOs had higher repellency effects against the adults of T. castaneum than 1% of concentration after 30 min of exposure (Table 1). On the other hand, all tested concentrations of S. aromaticum oil were more repellent than M. chamomilla after 30 min of exposure. Generally, the average numbers of repelled T. castaneum adults after 180 minutes were significantly higher than those after 30 minutes of exposure with all concentrations of P. dulcis, L. officinalis, and E. camaldulensis EOs. The EOs of C. lemon had significant diffferences in repellent effects among the exposure times at 5, 10, and 15% concentrations (Table 1). The 10 and 15% concentrations of S. chinensis and M. chamomilla EOs had a significantly diffferent repellency effects between the exposure times. Also, the repellency effect was not significantly varied in each concentration with the exposure time in S. aromaticum and A. sativum EOs.

Comment 4: There are many format and grammatical errors in the manuscript please modify the full manuscript.

Response 4: Thank you very much for your advice; we corrected the manuscript according to your advice. 

Comment 5: Which paragraph of text corresponds to your picture 1? How was the data in Table 2 obtained?

Response 5: Thank you very much for your comment we added the Fig1 in the following sentence which described figure 1. As shown in Fig. (1), 15 % concentrations of P. dulcis and M. chamomilla EOs have higher mortality rates of T. castaneum than those of S. aromaticum, A. sativum, E. camaldulensis, L. officinalis, and S. chinensis after 24 hrs of contact test. Also, C. lemon EO had a higher mortality rate than those of S. aromaticum, A. sativum, E. camaldulensis, and L. officinalis in 15% of concentration. The mortality percentages of P. dulcis, M. chamomilla, and C. lemon were 81.33, 81.33, and 77.33%, respectively, in 15% concentration. The mortality rates were significantly higher in 10 and 15% concentrations compared with 1 and 5% concentrations of P. dulcis, M. chamomilla, and C. lemon EOs. So, the mortality percentages in some of the tested EOs are increased by increasing the EOs concentration.

How was the data in Table 2 obtained:

Thank you very much for your comment, we added the following sentence according to your advice: The values of LC50 and LC90 as well as their confidence limits were conducted by using probit analysis by SPSS software [28] according to Finney's method [29].    

Reviewer 3 Report

Review MS: Biology-1483040

Present study investigates the repellent and toxicant effect of some essential oils (EOs) against the adult of the serious stored product pest, Tribolium castaneum. At first view, the manuscript is well written and the applied methods to investigate the hypothesis are good, they connect well with other studies in this research field. At the same time the manuscript has deficiency in every section from the introduction to the discussion, which must be corrected/completed/improved to became accepted for publication. The greatest question is the novelty of the study. It is not highlighted at all, in what aspect provide the study new results for researchers is this field? Repellency and toxicity were already tested at all the listed EOs in previous studies; maybe one advantage that these EOs present here together. However, the discussion contains some comparisons with previous results in this subject, according to the introduction reader could believe that these EOs were tested here first. Even so, this study is valuable and could be published, however only after major revision. Detailed comments could be found below.

SIMPLE SUMMARY – scientific names should be in italics!

ABSTRACT – according to the journal’s author guideline, abstract should not exceed 200 words. In the present form, it contains 303 words, so it should be shortened.

The INTRODUCTION provides a clear and concise presentation of the plant protection problems associated with stored products and the applicability of essential oils (EOs) in this area. At the same time, small review about the knowledge of Tribolium castaneum and EO is lacking, as one of the most studied insects in this regard is T. castaneum (based for example on one of the references cited by MS).

Also, the aim of the study is missing. All the listed EOs were already tested to T. castaneum in different studies. What is the novelty of your study except that you test these EOs in a common study which make their repellency and toxicity effect easy to compare? What was the reason to select these EOs and not others? This information would increase the quality of the introduction.

            Why the EOs were tested only on adults, which is the most susceptible life stage in connection with EOs. If the aim of the study is to find the best EOs for practical application, it would be important to test them on other life stages (e.g., larvae) and behaviours (oviposition, egg hatching, etc.).

MATERIAL AND METHODS

The selected bioassays are adequate and well accepted in this area, however the evaluation of the data could be improved.

Repellency assay - Other studies of this area usually does not count directly with the number of insects on the treated or control area, but calculate the percentage repellency, which are compared in a subsequent analysis. According to the Table 1. percentage repellency was calculated, so please indicate this in the M&M as well.

Statistical analysis – ANOVA has some assumption whish should be tested before the analysis. Did you check them? If yes, please indicate here as well as the transformation method if it was needed.

RESULTS

Repellency assay

            what does “botanical origin” mean in the text? Neither in the introduction, nor in the discussion it is not explained.

            Why P. dulcis has moderate repellency and M. chamomilla weak? According to the applied statistic, there are not significant differences among them neither in the concentrations. However, the number of repelled adults is numerically different, the value of their percentage of repellency are the same (according to the table I).

            Regarding to the last paragraph of this section: is 80% of repellent effect as bad? Yes, it is a lower value that that of the S. aromaticum, but in general good values (according to the repellency classes of botanical oils)

Comparing the average number of insects on the control area among all the concentrations and type of EOs is not correct. It means too many comparisons even in case of Tukey’s post-hoc test, which is corrected for Type I error. So, you should restrict the number of comparisons or aggregate your data according to concrete questions: e.g., differences in repellency of the given EOs among concentrations or differences in repellency among EOs on the given concentrations. Better to use a two-way ANOVA (EOs and concentration as factors) or (as concentration is a continuous variable) an ANCOVA (where concentration is a covariant and EOs a factor), so that you can investigate the effect of the different variables and exclude unnecessary comparisons.

Style of the table should be improved (e.g., by using one type of data – percent repellency or average No. of insect on untreated area), however, instead of a table you should use a connected scatterplot, where X-axis is the concentration, Y-axis is the mean percentage repellency and the grouping factor is the EOs.

Insect mortality bioassay

            Why had S. aromaticum high performance in this investigation if it has high LD values and low mortality to the T. castaneum?

            Exact mortality values should be inserted in the last paragraph of this section.

DISCUSSION

            General paragraphs of the discussion of a scientific manuscript are missing - e.g., importance, limitations of the study, future direction of the research -, so in this regard it should be rethink and redefine.

            Repellent effect of S. aromaticum and A. sativum are discussed and compared with other studies, however there are no conclusions in connection with the other tested EOs. Also, the part in connection with mortality is incomplete in this regard.

Sentence in connection with the repellent effect of M. chamomilla is not explained. While in the referred study, this EO had a high repellency, in the present one it had the lowest. What could be the reason of this difference?

            Novelty and significance of the presented research is not highlighted at all, it should be completed.

CONCLUSION

According to the journal’s author guide, conclusion is not mandatory, but it helps to summarize the discussion if it is unusually long or complex. So, to my opinion this section could be omitted, however, the paragraph could be attached to the end of discussion with some completion about the concrete way of application of these EOs.

Author Response

Reviewer 3, comments and Suggestions for Authors

The authors would like to thank you for your valuable comments. It was your valuable and insightful comments that led to possible improvements in the current version. All corrections were done according to your advice.

Comment 1:  SIMPLE SUMMARY – scientific names should be in italics!

Response 1: Thank you very much for your comment. We changed all scientific names and become italic.

Comment 2: ABSTRACT – according to the journal’s author guideline, abstract should not exceed 200 words. In the present form, it contains 303 words, so it should be shortened.

Response 2: Thank you very much for your advice we changed the abstract and it becomes 194 words with clear information.

Comment 3: The INTRODUCTION 

Response 3: Thank you very much for your advice, comments and suggestions in the introduction part and we changed it according to your comments.  

Comment 3-1: provide a clear and concise presentation of the plant protection problems associated with stored products and the applicability of essential oils (EOs) in this area. At the same time, small review about the knowledge of Tribolium castaneum and EO is lacking, as one of the most studied insects in this regard is T. castaneum (based for example on one of the references cited by MS).

Response 3-1: Thank you very much for your advice, comment, we added more clear information and new references according to your advice: Previous research indicated that EOs of four spice plants and their main components have clear toxicity and repellant actions against T. castaneum and Lasioderma serricorne [19]. Jahromi et al. [19] indicated that at high concentrations, the natural garlic emulsion has the greatest repellency impact against T. castaneum. Also, Oryzaephilus surinamensis was completely died by the essential oil of M. chamomilla at concentrations greater than 0.5 percent [20]. At a concentration of 0.5 percent, lemongrass essential oil was found to be extremely repellent to Acanthoscelides obtectus and T. castaneum as indicated by Gvozdenac et al. [21].

Also, the aim of the study is missing. All the listed EOs were already tested to T. castaneum in different studies. What is the novelty of your study except that you test these EOs in a common study which make their repellency and toxicity effect easy to compare? What was the reason to select these EOs and not others? This information would increase the quality of the introduction.

Thank you very much for your advice and comments about the novelty of this study. We clarified this point according to your advice in the following sentence.  The main aim of this study was to compare the repellency and toxicity effects of eight EOs against adults of T. castaneum Herbst.

Thank you very much for your advice about the reason to select these EOs and not others. We explained why we chose the tested EOs according to your advice:  In recent years, several previous researchers have reported the repellency or mortality effects of Syzygium aromaticum, Allium sativum, Eucalyptus camaldulensis, Lavandula officinalis, Simmondsia chinensis, Matricaria chamomilla, Citrus lemon, and Prunus dulcis EOs, against Tribolium castaneum. However detailed information about the comparison between the repellency and toxicity effects are still lacking. So, the main aim of this study was to compare between repellency and toxicity effects of the tested EOs to determine the best application.

 Why the EOs were tested only on adults, which is the most susceptible life stage in connection with EOs. If the aim of the study is to find the best EOs for practical application, it would be important to test them on other life stages (e.g., larvae) and behaviours (oviposition, egg hatching, etc.).

Thank you very much for your advices and comments and we focused during this study on the adults to control it as the most important stage before laying eggs. Also, we will do these points in the nearest future and we added the following sentence in the manuscript In the present research (2021), we could not estimate the comparison between the repellency and toxicity effects of tested EOs against other stages of T. castaneum Herbst including egg, larvae, and pupae. So, we would like to do these points in the nearest future.

Comment 4: MATERIAL AND METHODS

The selected bioassays are adequate and well accepted in this area, however the evaluation of the data could be improved.

Response 4: Thank you very much for your advices and comments: We rechecked the data and we determined the differences in each exposure time. And the comparison between the exposure times for each concentration was done by one way ANOVA as shown in the following paragraph.

The experimental data were analyzed using SPSS software [29]. The Shapiro–Wilk normality test was used to test the normality of the data, which indicated the normal distribution of the data. Therefore, the analysis was performed on the original data. A two-way analysis of variance (ANOVA)was used to find out the repellency and mortality differences among the tested EOs with different concentrations, against T. castaneum, and Tukey's test was used to find out the variations among the means for each exposure time (the EOs and concentration were used as fixed factors). In addition, one-way ANOVA was used to conduct the repellency differences for each concentration between 30 min and 180 min of the exposure time.

Repellency assay - Other studies of this area usually does not count directly with the number of insects on the treated or control area, but calculate the percentage repellency, which are compared in a subsequent analysis. According to the Table 1 percentage repellency was calculated, so please indicate this in the M&M as well.

Thank you very much for your recommendation and we added the percentages of repellency and we deleted the numbers, according to your advice. The percentages were calculated: this equation [27]:

PR = [(Nc – Nt)/(Nc + Nt)] × 100.

Where PR means the repellency percentages after exposure time; Nc means the insect numbers on the untreated area after the exposure; Nt means the insect numbers on the treated area after the exposure.

Statistical analysis – ANOVA has some assumption whish should be tested before the analysis. Did you check them? If yes, please indicate here as well as the transformation method if it was needed.

Thank you very much for your recommendation, we checked the normality by SPSS, descriptive statistics, explore. We found the data are in the normal distribution. We did the statistical according to your recommend.  The experimental data were analyzed using SPSS software [29]. The Shapiro–Wilk normality test was used to test the normality of the data, which indicated the normal distribution of the data. Therefore, the analysis was performed on the original data. A two-way analysis of variance (ANOVA)was used to find out the repellency and mortality differences among the tested EOs with different concentrations, against T. castaneum, and Tukey's test was used to find out the variations among the means for each exposure time (the EOs and concentration were used as fixed factors). In addition, one-way ANOVA was used to conduct the repellency differences for each concentration between 30 min and 180 min of the exposure time.

Comment 5: RESULTS

 Repellency assay

Comment: What does “botanical origin” mean in the text? Neither in the introduction, nor in the discussion it is not explained.

Response: Thank you very much for your great comment and we meant the plant species and it was a mistake and we changed it according to your advice.

           Why P. dulcis has moderate repellency and M. chamomilla weak? According to the applied statistic, there are not significant differences among them neither in the concentrations. However, the number of repelled adults is numerically different, the value of their percentage of repellency are the same (according to the table I).

Response: Thank you very much for your comments, we rechecked all the significant differences and we added in the text as shown in the following paragraph according to your comments.

The repellency effect of 10 and 15% (v/v) of S. aromaticum oil on T. castaneum was significantly higher than E. camaldulensis, L. officinalis, S. chinensis, M. chamomilla, C. lemon, and P. dulcis (Table 1). Also, the oil of A. sativum at 15% concentration had a higher significant repellency effect compared with E. camaldulensis, L. officinalis, S. chinensis, and M. chamomilla. The 15% of S. aromaticum, P. dulcis, E. camaldulensis, and C. lemon EOs had higher repellency effects against the adults of T. castaneum than 1% of concentration after 30 min of exposure (Table 1). On the other hand, all tested concentrations of S. aromaticum oil were more repellent than M. chamomilla after 30 min of exposure

Generally, the average numbers of repelled T. castaneum adults after 180 minutes were significantly higher than after 30 minutes of exposure with all dilution percentages of Prunus dulcis, L. officinalis and E. camaldulensis EOs. The essential oils of Citrus lemon had a significant repellent effect between the exposure times at 5, 10 and 15% dilution (Table 1). The 10 and 15% dilution percentages of Simmondsia chinensis and M. chamomilla EOs had a significant repellency effect between the exposure times. Also, the repellency effect was not significantly differed in each concentration with the exposure time in Syzygium aromaticum and garlic Allium sativum EOs.

            Regarding to the last paragraph of this section: is 80% of repellent effect as bad? Yes, it is a lower value that that of the S. aromaticum, but in general good values (according to the repellency classes of botanical oils)

Response: Thank you very much for your valuable comments and we rechecked the manuscript. We corrected this point according the differences in Table. And the corrected values were inserted in the manuscript according to the differences in Table 1, according to your advice.

Generally, the average numbers of repelled T. castaneum adults after 180 minutes were significantly higher than after 30 minutes of exposure with all dilution percentages of Prunus dulcis, L. officinalis and E. camaldulensis EOs. The essential oils of Citrus lemon had a significant repellent effect between the exposure times at 5, 10 and 15% dilution (Table 1). The 10 and 15% dilution percentages of Simmondsia chinensis and M. chamomilla EOs had a significant repellency effect between the exposure times. Also, the repellency effect was not significantly varied in each concentration with the exposure time in Syzygium aromaticum and garlic Allium sativum EOs.

Comparing the average number of insects on the control area among all the concentrations and type of EOs is not correct. It means too many comparisons even in case of Tukey’s post-hoc test, which is corrected for Type I error. So, you should restrict the number of comparisons or aggregate your data according to concrete questions: e.g., differences in repellency of the given EOs among concentrations or differences in repellency among EOs on the given concentrations. Better to use a two-way ANOVA (EOs and concentration as factors) or (as concentration is a continuous variable) an ANCOVA (where concentration is a covariant and EOs a factor), so that you can investigate the effect of the different variables and exclude unnecessary comparisons.

Response: Thank you very much for your great comment and we rechecked the data according to your advice and we added the corrections according to your recommendation:  The repellency effect of 10 and 15% (v/v) of S. aromaticum oil on T. castaneum was significantly higher than E. camaldulensis, L. officinalis, S. chinensis, M. chamomilla, C. lemon, and P. dulcis (Table 1). Also, the oil of A. sativum at 15% concentration had a higher significant repellency effect compared with E. camaldulensis, L. officinalis, S. chinensis, and M. chamomilla. The 15% of S. aromaticum, P. dulcis, E. camaldulensis, and C. lemon EOs had higher repellency effects against the adults of T. castaneum than 1% of concentration after 30 min of exposure (Table 1). On the other hand, all tested concentrations of S. aromaticum oil were more repellent than M. chamomilla after 30 min of exposure

Style of the table should be improved (e.g., by using one type of data – percent repellency or average No. of insect on untreated area), however, instead of a table you should use a connected scatterplot, where X-axis is the concentration, Y-axis is the mean percentage repellency and the grouping factor is the EOs.

Response: Thank you very much for your great comment:  We revised that point according to your comment and we added the repelled percentages of T. castanium in the Table 1 and we added a new column to show the significant between the two exposure times for each concentration.

Insect mortality bioassay

            Why had S. aromaticum high performance in this investigation if it has high LD values and low mortality to the T. castaneum?

Response: Thank you very much for your comments and we added the following sentence to describe the reasons for that point: We suppose the reason for the potent repellent effect of S. aromaticum maybe because of the odour, but with less mortality percentage by direct contact may be due to the contact toxicity depending on the attached chemical substances to the insect body and it may have low toxicant components.

            Exact mortality values should be inserted in the last paragraph of this section.

Response: Thank you very much for your comments and we added the exact mortality values in this sentence according to your recommendation: As shown in Fig. (1), 15 percentages of P. dulcis and M. chamomilla EOs had higher mortality rates of T. castaneum than S. aromaticum, A. sativum, E. camaldulensis, L. officinalis and S. chinensis after 24 hrs of contact test. Also, C. lemon EO had a higher mortality rate than S. aromaticum, A. sativum, E. camaldulensis and L. officinalis in 15% of dilution percentage. The mortality percentages of P. dulcis, M. chamomilla, and C. lemon were 81.33, 81.33 and 77.33% respectively, in 15% of dilution percentage. The mortality rates were significantly higher in 10 and 15% compared with 1 and 5% dilution percentages with P. dulcis, M. chamomilla and C. lemon EOs. So, the mortality percentages in some of the tested EOs are increased by increasing the EOs percentage.

Comment 6: DISCUSSION

            General paragraphs of the discussion of a scientific manuscript are missing - e.g., importance, limitations of the study, future direction of the research -, so in this regard it should be rethink and redefine.

Response: Thank you very much for your advice and we added this sentence according to your recommendation: Recently, many researchers have evaluated the repellency or mortality effects for the tested EOs. However detailed results about the comparison between the repellent and insecticidal effects are still insufficient. Thus, this study was conducted to find out the differences between repellency and toxicity effects of the tested EOs.

            Repellent effect of S. aromaticum and A. sativum are discussed and compared with other studies, however there are no conclusions in connection with the other tested EOs. Also, the part in connection with mortality is incomplete in this regard.

Response: thank you very much for your recommendation. We did the changes according to your comment: In this study, the repellence tests indicated that S. aromaticum and A. sativum EOs had good potential as repellent substances against T. castaneum after 30 min of exposure. The effectiveness of S. aromaticum and A. sativum EOs could be due to some factors which contribute to enhance their repellent activity such as their odour, chemical compositions and main components, as shown in previous works such as, Abo-El-Saad [31] showed that the significant insecticidal activity of clove oil might be attributed to the main components, eugenol and β-caryophyllene. The high repellent activity of S. aromaticum EO may be due to eugenol because it is the main constituent of clove oil (48.92%) as determined by GC-MS [31] also, Boraei [32] reported that the main component of clove EO was eugenol (37.43%) as analyzed by GC-MS. In many previous studies, this component has been indicated to be a strong repellent compound against many insect pests [33,34]. Also, the results indicated that the 15% of Allium sativum EO was higher in repellency effect than E. camaldulensis, L. officinalis, S. chinensis and M. chamomilla. These findings are in harmony with those reported by Jahromi et al. [19] they have shown that the natural garlic emulsion, A. sativum had the highest repellency effect against T. castaneum at the high concentration. In addition, the T. castaneum was more susceptible to garlic emulsion than Lasioderma serricorne [19]. Rahman and Motoyama [35] found that the active volatile components are sulfide components made by the rapid degradation of allicin. Rahman and Schmidt [36] have studied the chemical analysis of garlic extract by GC-MS and have indicated that allicin was a main component and had repellency impacts on some insect pests. Our results showed that 1 % of M. chamomilla EO repelled 15.33±0.33 individuals of T. castaneum after 3 hrs of exposure while Al-Jabr [20] has shown that the repellent effect of M. chamomilla against T. castaneum was 84.73% after 48 hrs at 1% concentration. The differences between our findings and Al-Jabr [20] may be because Al-Jabr [20] have used a long exposure time of exposure time 48 hrs. We suppose the reason of potent repellent effect of S. aromaticum may be because of their odour,  but with less mortality percentage by direct contact may be due to the contact toxicity depending on the attached chemical substances to the insect body and it may have low toxicant components. These results may recommend that using S. aromaticum and A. sativum as a repellent EO might be a promising method to control T. castaneum.

The results indicated that 15% of P. dulcis and M. chamomilla were more effective by contact toxicity against T. castaneum than S. aromaticum, A. sativum, E. camaldulensis, L. officinalis and S. chinensis. Also, 15% of the tested EOs, C. lemon EO had a higher mortality rate than S. aromaticum, A. sativum, E. camaldulensis and L. officinalis. The effectiveness of P. dulcis and M. chamomilla are in harmony with those reported by Al-Jabr [20] who showed that 1% of P. amygdalus was more effective against T. castaneum with complete mortality after 14 days of exposure compared with Cinnamomum camphora, Cymbopogon winterianus, M. chamomilla, Mentha viridis, P. amygdalus var amara, Rosmarinus afficinalis, and S. chinensis. Azab et al. [37] have shown that the LC50 and LC90 of sweet almond EO to adults of Oryzaephilus surinamensis were 4.52% and 5.55% (v/w), respectively after 7 days of exposure time and their mortality percentages are enhanced by increasing the EOs percentage and the period of exposure. Also, Matsumoto et al. [38] have shown that the essential oils of bitter almond, spearmint and birch bark were used in a composition that was sold as a pesticide, insect repellent and acaricide. Al-Jabr [20] has indicated that M. chamomilla EO at a concentration of more than 0.5% had a complete mortality effect against O. surinamensis. Also, Padin et al. [39] have shown that the methanolic extracts of M. chamomilla had a 57% mortality rate against T. castaneum after 7 days. El-Bakry et al. [40] have shown that the LC50 value of C. sinensis was 35 µl/L and it was the most effective one against T. castaneum.

Sentence in connection with the repellent effect of M. chamomilla is not explained. While in the referred study, this EO had a high repellency, in the present one it had the lowest. What could be the reason of this difference?

Response: Thank you very much for your comment we added our suggestion for the differences according to your advice: Our results showed that 1 % of M. chamomilla EO repelled 15.33±0.33 individuals of T. castaneum after 3 hrs of exposure while Al-Jabr [20] has shown that the repellent effect of M. chamomilla against T. castaneum was 84.73% after 48 hrs at 1% concentration. The differences between our findings and Al-Jabr [20] maybe because Al-Jabr [20] have used a long exposure time 48 hrs. These results may recommend that using S. aromaticum and A. sativum as a repellent EO might be a promising method to control T. castaneum.

            Novelty and significance of the presented research is not highlighted at all, it should be completed.

Response 6: Thank you very much for your comment and we added the novelty and the differences in all parts of the manuscript like the following paragraph: This study was conducted to compare the repellent effect and contact toxicity of eight essential oils (EOs) included Syzygium aromaticum, Allium sativum, Eucalyptus camaldulensis, Lavandula officinalis, Simmondsia chinensis, Matricaria chamomilla, Citrus lemon, and Prunus dulcis against adults of Tribolium castaneum Herbst. Four concentrations (1, 5, 10, and 15% in acetone solvent) of each EO were tested. The 10 and 15% concentrations of S. aromaticum EO had a high repellency effect against T. castaneum compared with E. camaldulensis, L. officinalis, S. chinensis, M. chamomilla, C. lemon, and P. dulcis. The 15 % concentration of P. dulcis and M. chamomilla EOs have a significant impact on the mortality rate of T. castaneum compared with S. aromaticum, A. sativum, E. camaldulensis, L. officinalis, and S. chinensis after 24 hrs of contact test. Also, the 15% concentration of C. lemon EO caused a greater mortality percentage compared with S. aromaticum, A. sativum, E. camaldulensis, and L. officinalis. It could be concluded that using the S. aromaticum EO as a repellent oil, and for contact toxicity using P. dulcis, M. chamomilla, and C. lemon to treat the flour infested by T. castaneum can play an important role in protecting stored grains and their products.

Comment 7: CONCLUSION: According to the journal’s author guide, conclusion is not mandatory, but it helps to summarize the discussion if it is unusually long or complex. So, to my opinion this section could be omitted, however, the paragraph could be attached to the end of discussion with some completion about the concrete way of application of these EOs.

Response 7: Thank you very much for your comment, we attached the conclusion at the end of the following paragraph: Generally, the major components of plant EOs are many monoterpenoids like, d-limonene, Æ-terpineol, â-myrcene, pulegone, and linalool which affect negatively on many insect pests such as the German cockroach and the house fly [22,23], linalool was considered as an acetylcholinesterase inhibitor and it has been demonstrated a potent contributor to the repellent and insecticidal activities [25,24]. Bhavaniramya et al. [40] have indicated that the use of EOs, clove, lemon and thyme cinnamon has increased the storage periods and have kept the good quality of food safety. Plant EOs are generally utilized in food products for food preservation because of their odour, tastes, and strong antibacterial properties. For instance, the EOs of citrus including monoterpenes, sesquiterpenes, and oxygenated derivatives have strong inhibitory effects against harmful bacteria, proposing that they could be used as flavouring and antioxidants preservative materials of food [40]. Almost EOs contain terpenes and aromatic volatile chemicals, which play a significant role in food safety without decreasing quality [40]. Thus, the use of EOs can protect the food products for a long time without decreasing the quality. It could be concluded that this study may suggest that using S. aromaticum as a repellent material and for contact mortality using P. dulcis, C. lemon and M. chamomilla EOs might be useful for the management of T. castaneum to protect stored grains and their products.

Round 2

Reviewer 2 Report

This article has been revised to have a greater improvement, but there are still some problems, it is recommended to accept after minor revision, specific problems are as follows:

Although acetone may volatilize, whether the volatilization is sufficient or not is still in doubt. I hope that the author can provide the relevant data after the volatile residue, otherwise, I still have doubts about the safety of this method.

Author Response

The authors would like to thank you for your valuable and great comments. It was your valuable and insightful comments that led to possible improvements in the current version. The corrections were done according to your advice.

Comment 1: Although acetone may volatilize, whether the volatilization is sufficient or not is still in doubt. I hope that the author can provide the relevant data after the volatile residue, otherwise, I still have doubts about the safety of this method.

Response 1:  We would like to thank you very much for your comment because it is very important to know the safety of acetone in this study and thank you for bringing it to my attention: however we could not evaluate the acetone residues in our study because we found many previous papers used it as a solvent but we will do it in the near future and we indicated that in the manuscript as shown in the following sentence. Also, in the future, we would like to evaluate the acetone residues on treated stored grains and their products to identify the level of safety

Reviewer 3 Report

Thanks to the authors for taking into consideration my comments on the manuscript and for correcting and completing it where they agreed with my opinion. The quality of the manuscript has definitely improved, it is more suitable for publication than the first version. However, there is one point where I think the manuscript still needs to be improved, and that is the statistics of the repellency tests.  My comments in connection with this point can be found in the attached file. Correct evaluation of the results not only makes it easier for the reader to compare the results with other works, but also makes it easier for researchers to draw the right conclusions.

Author Response

The authors would like to thank you for your valuable and great comments. It was your valuable and insightful comments that led to possible improvements in the current version. The corrections were done according to your advice.

Comment 1: Although acetone may volatilize, whether the volatilization is sufficient or not is still in doubt. I hope that the author can provide the relevant data after the volatile residue, otherwise, I still have doubts about the safety of this method.

Response 1:  We would like to thank you very much for your comment because it is very important to know the safety of acetone in this study and thank you for bringing it to my attention: however we could not evaluate the acetone residues in our study because we found many previous papers used it as a solvent but we will do it in the near future and we indicated that in the manuscript as shown in the following sentence. Also, in the future, we would like to evaluate the acetone residues on treated stored grains and their products to identify the level of safety.

.****************************************************************************************************************************************************************************************************

Reviewer 3, comments and Suggestions for Authors

The authors would like to thank you very much for your valuable comments and your advice. It was your valuable and insightful comments that led to possible improvements in the current version. All corrections were done according to your comments.

Comment 1: If you conducted ANOVA, than please provide the main results of ANOVA (F, df, P) in connection with the factors and interactions as well. A I wrote earlier, it is not correct to compare everything with everything, it is too many comparisons. You should restrict the number of comparisons or aggregate your data according to concrete questions: e.g., differences in repellency of the given EOs among concentrations or differences in repellency among EOs on the given concentrations. Please check the paper Jahromi et al 2014 (19th of your reference list), it contains a correct statistical evaluation for repellency data. Also it is worth to read Yang et al (Yang, K. et al. 2014, J of Asia-specific Entomology, 17(3):459-466), who built a GLM to their data, where componds, concentrations and times were evaluated at once.

Response 1:  We would like to thank you very much for your comments and your advice we checked all our data and we conducted the statistical analysis according to your advice: The experimental data were analyzed using SPSS software [29]. The Shapiro–Wilk normality test was used to test the normality of the data, which indicated the normal distribution of the data. Therefore, the analysis was performed on the original data. Data were used in univariate analysis using SPSS software [29]. The statistical analysis of data was conducted on each dependent variable and the experimental treatments were compared for significant differences with a two way ANOVA and the differences between the means were estimated using Tukey’s test.

Furthermore, Table (2) shows that the F values of the concentrations were statistically significant for S. aromaticum, A. sativum, P. dulcis, L. officinalis, E. camaldulensis, Citrus lemon and M. chamomilla EOs. The F values of the exposure time were statistically significant for the EOs of S. aromaticum, P. dulcis, L. officinalis, S. chinensis, E. camaldulensis, C. lemon and M. chamomilla. Also, the F values of the interaction between concentration and exposure time were significant for S. aromaticum, A. sativum, C. lemon  and M. chamomilla EOs (Table 2).

Table 1: Repellency impact of eight essential oils against T. castaneum after 30 and 180 minutes of exposure.

Essential oils

Repelled adults (%) after 30 min of exposure

Repelled adults (%) after 180 min of exposure

Concentrations

Concentrations

1%

5%

10%

15%

1%

5%

10%

15%

S. aromaticum

64.44 a

77.77a

82.22 a

95.55 a

82.22 a

86.66 a

95.55 a

100 a

A. sativum

51.11ab

51.11 b

60 b

73.33 b

55.55 bc

60 b

64.44 b

91.11 ab

S. chinensis

40 bc

48.88 b

31.11 cd

31.11 de

60 bc

64.44 b

73.33 b

73.33 c

P. dulcis

22.22 cd

44.44 b

53.33 b

57.77 bc

42.22 c

64.44 b

82.22 ab

91.11 ab

L. officinalis

17.77 d

31.11 bc

48.88 bc

48.88 cd

73.33 ab

73.33 ab

77.77 ab

82.22 bc

E. camaldulensis

8.88 d

26.66 bc

44.44 bc

44.44 cd

60 bc

64.44 b

64.44 b

77.77 bc

C. lemon

6.66 d

22.22 bc

57.77 b

31.11 de

2.22 d

55.55 b

77.77ab

82.22 bc

M. chamomilla

2.22 d

4.44 c

22.22 d

22.22 e

0.00 d

15.55 c

46.66 c

66.66 c

Table 2: Analysis of variance of different EOs of the tested insect's repellency.

Essential oils

Concentrations

Exposure time

Interaction between concentrations  and exposure time

df

Mean square

F

Sig.

df

Mean square

F

Sig.

df

Mean square

F

Sig.

S. aromaticum

3

2200.000

17.471

00.000**

1

1896.296

15.059

00.001**

3

444.444

3.529

00.039*

A. sativum

3

1037.037

8.235

00.002**

1

474.07

3.76

00.070 NS

3

59.259

.471

00.707 NS

S. chinensis

3

46.914

00.204

00.892 NS

1

5400.000

23.516

00.000**

3

46.914

.204

00.892 NS

P. dulcis

3

2081.481

17.562

00.000**

1

3918.519

33.062

00.000**

3

66.667

.563

00.647 NS

L. officinalis

3

533.333

4.000

00.027*

1

9600.000

72.000

00.000**

3

207.407

1.556

00.239 NS

E. camaldulensis

3

800.000

9.000

00.001**

1

7585.185

85.333

00.000**

3

246.914

2.778

00.075 NS

C. lemon

3

4717.901

32.249

00.000**

1

3750.000

25.633

00.000**

3

703.086

4.806

00.014*

M. chamomilla

3

2200.000

17.471

00.000**

1

1896.296

15.059

00.001**

3

444.444

3.529

00.039*

Comment 2: However, ANOVA give the same result, a simple t-test is enough here.

Response 2:  We would like to thank you very much for your advice we checked that point according to your advice by univariate analysis as shown in Table 1.

Commented [.3]: Where are SDs?

Response 2:  We would like to thank you very much for your comment. We rechecked the data and we deleted SD because it is difficult to add it in that table because of many data with the new style as you recommended.

Commented [.4]: It is important to indicate here, what kind of statistical test these values come from?

Response 4:  We would like to thank you very much for your advice and we added the used test as shown in the following text. Values are the mean percentages of repelled insects. The means of each column followed by the same letter do not differ significantly P > 0.05 as determined by Tukey's test.

Commented [.5]: How do you mean this? S. aromaticum had the second wrost result according to the table 2.

Response 5:  We would like to thank you for that comment. It was a mistake and we changes it with P. dulcis.

Commented [.6]: It is true even in case of EOs with lower mortality rate.

Response 6:  We would like to thank you for that advice and the text was changed according to your advice to the following text. So, the mortality percentages of T. castaneum are increased by increasing the EOs concentration.

Commented [.7]: It is important to indicate here informations about the signs on the figure. What are labels above the columns means? Which statistics did you use?

Response 7:  We would like to thank you for that comment and we added the following sentence below the Fig1. The mortality percentages of each concentration followed by the same letter are not significantly different at the 0.05 as determined by Tukey's test.

Commented [.8]: repellency ratio is a good index to compare the repellency, so it would be better to write here PR%, that you could compare directly the referred study.

Response 8:  We would like to thank you for that advice and we changed the mortality number to PR. 2.22%.

Commented [.9]: It is not clear what do you mean here, please detail it.

Response 9: thank you very much for your advice and changed the sentence to be more suitable according to your advice. We suppose the reason for the low mortality percentage of S. aromaticum by direct contact may be due to its low toxicant components which are attached to the insect body. These results may recommend that using S. aromaticum as a repellent EO and P. dulcis and M. chamomilla EOs for a contact toxicity might be a promising method to control T. castaneum.

Commented [.10]: Better fit to the final conclusion

Response 10:  We would like to thank you for that advice and changed that sentence to be more suitable according to your recommendation. These results may recommend that using S. aromaticum as a repellent EO and P. dulcis and M. chamomilla EOs for a contact toxicity might be a promising method to control T. castaneum.

Commented [.11]: concentration?

Response 11:  We would like to thank you for that recommendation. Yes, it is concentration.

Finally thank you very much for your comments. And all the other corrections were done according to your advice.